# Genetic Relationships of 118 *Castanea* Specific Germplasms and Construction of Their Molecular ID Based on Morphological Characteristics and SSR Markers

**DOI:** 10.3390/plants12071438

**Published:** 2023-03-24

**Authors:** Xiaoqian Bai, Shijie Zhang, Wu Wang, Yu Chen, Yuqiang Zhao, Fenghou Shi, Cancan Zhu

**Affiliations:** 1Jiangsu Key Laboratory for the Research and Utilization of Plant Resources, Institute of Botany, Jiangsu Province and Chinese Academy of Sciences (Nanjing Botanical Garden Mem. Sun Yat-Sen), Nanjing 210014, China; 2College of Forestry, Nanjing Forestry University, Nanjing 210037, China

**Keywords:** *Castanea*, phenotypic traits, variety identification, genetic diversity, molecular identity card, population structure

## Abstract

To understand the genetic relationships of *Castanea* species, 16 phenotypic traits were measured, simple sequence repeat (SSR) markers were analyzed, and molecular identity cards (IDs) were constructed for 118 *Castanea* materials using fluorescent capillary electrophoresis. The coefficient of variation values of the 16 morphological traits of the test materials ranged from 11.11% to 60.38%. A total of 58 alleles were detected using six pairs of SSR core primers, with an average number of 9.7 alleles per locus. The average number of valid alleles per locus was 3.9419 and the proportion of valid alleles was 40.78%. A total of 105 genotypes were detected, and the number of genotypic species that could be amplified per primer pair ranged from 8 to 26. The mean value of the observed heterozygosity was 0.4986. The variation in the *He*, *H*, and *PIC* values was similar; the size of *I* value was approximately 2.21 times larger, and its mean number of variations was 0.7390, 0.7359, 0.6985, and 1.6015, respectively. The classification of 118 *Castanea* species was performed using three analytical methods: structure analysis, neighbor-joining (NJ) cluster analysis, and principal coordinate analysis (PCoA), and the results of the three methods were in high agreement. Six pairs of SSR core primers with high polymorphism and strong discriminatory properties were used to identify 118 *Castanea* plants, and a unique molecular ID card was constructed for each material. These results provide insight into the genetic diversity and population structure of *Castanea* plants and a theoretical basis for improving the phenomenon of mixed varieties and substandard plants in the *Castanea* plant market.

## 1. Introduction

The *Castanea*, which is cross-pollinated, includes seven major species: *Castanea mollissima* Bl., *Castanea henryi* (Skam) Rehd. et Wils., *Castanea seguinii* Dode, *Castanea crenata* S. et Z., *Castanea dentata* (Marsh.) Brokh, *Castanea sativa* Mill., and *Castanea pumila* Mill. Among them, the Chinese chestnut (*C. mollissima* Bl.) exhibits superior disease resistance traits, such as exhibiting resistance to two major pathogens, *Cryphonectria parasitica* and *Phytophthora cinnamomic* [1]. Chinese chestnut has very high commercial economic value in terms of disease resistance and fruit quality. As a result, it may be a useful resource in the selection and breeding of resistant *Castanea* plants and has a crucial role. *Castanea* has comparatively high nutritional value due to its essential amino acids, fatty acids, vitamins, minerals, dietary fiber, and other nutrients. Additionally, cooked *Castanea* can strengthen the spleen, tone the kidneys, and nourish the stomach. It also has beneficial preventive and therapeutic effects on coronary heart disease, hypertension, and atherosclerosis [2,3]. However, there exist outstanding issues in the breeding, cultivation, and consumption of the *Castanea*, such as hybridization of varieties and homonyms; this has led to a rise in imitation and inferior varieties and caused losses to breeders, farmers, and consumers. Therefore, there is an urgent need to establish an accurate, rapid, and efficient method for breed identification.

With the rapid development of molecular biology, a variety of molecular marker technologies, such as randomly amplified polymorphic DNA (RAPD), amplified fragment length polymorphism (AFLP), simple sequence repeat (SSR) and single nucleotide polymorphism (SNP), have been widely used in the analysis of genetic diversity and plant variety identification. SSR molecular marker technology involves tandem repeat sequence consisting of 1–6 nucleotides as repeat units, which has excellent characteristics such as co-dominant inheritance, high polymorphism, good reproducibility, and stable and accurate results. It has emerged as a key molecular marker technology for plant genetic diversity analysis, population structure research, variety identification, and molecular ID construction [4,5,6,7,8].

The construction of various molecular ID cards is a highly efficient method for identifying germplasm resources and has been widely used for the identification and authentication of a variety of plants [9,10]. Molecular ID cards can not only digitize DNA molecular data but also combine molecular data with botanical taxonomic information, phenotypic trait information, selective breeding information, and other information. Molecular ID cards are also of great importance for germplasm resource management, which can carry out the fine-grained management of germplasm resources in a clear and systematic manner, and are conducive to organizing, conserving and managing germplasm resources. By generating QR code molecular ID cards, the molecular and phenotypic information of plants can be obtained through the act of scanning, which can contribute to more convenient, rapid, detailed, and comprehensive understanding of plant resources. The molecular ID cards are unique and can be identified by constructing unique ID cards for each species. Among the studies on *Castanea* plants, there are few reports on the construction of molecular ID cards for *Castanea* plants.

In this study, the genetic diversity and population structure of 118 materials from four species of *Castanea* were analyzed and systematically described using SSR molecular markers and fluorescence capillary electrophoresis; an exclusive molecular ID card was constructed for each material and used to identify the 118 distinct materials. The results of this study provide a molecular basis for understanding the genetic diversity and population structure of germplasm resources of *Castanea* and will serve as a reference for solving the problems of homonymy and synonymy.

## 2. Results

### 2.1. Phenotypic Traits and Morphological Analysis of Appearance

The coefficient of variation values of the 16 morphological traits of the test materials ranged from 11.11% to 60.38% (Table 1), among which the coefficient of variation values of the five traits, namely single cone weight, cone shell thickness, single nut weight, stigma length, and single kernel weight, were all higher than 30%. In addition, the coefficient of variation values of all morphological traits was in the range of 20%−30%, except for the minimum value of 11.11% for fruit shape index. Cluster analysis of phenotypic traits was performed based on the mean values of phenotypic trait data of the test materials (Appendix A), and the results of the analysis divided 118 *Castanea* plant materials into four groups (Figure 1), namely Taxon I, Taxon II, Taxon III, and Taxon IV. Taxon i included 21 materials, Taxon II included 38 materials, Taxon III included 31 materials, and Taxon IV included 28 materials.

The variation in morphological traits and appearance morphology of germplasm resources of different cultivars can be seen from the variogram of appearance morphology (Figure 2). For example, the size of cones and nuts of cultivar ‘Mao Pu’ was approximately twice as large as that of cultivar ‘Gui Hua Xiang’, and the length of spine of ‘Mao Pu’ was longer than that of ‘Gui Hua Xiang’. There are also differences in the color of the nut shells of different cultivars, the nut shell color of cultivar ‘Liu Yue Bao’ is black and more lusterless, the nut shell color of cultivar ‘Da Hong Pao’ is brownish red and the surface is smooth and shiny. It is worth noting that from the appearance of the nut shape, the cultivar ‘Jian Ding You Li’ has a teardrop-shaped nut, which is easier to distinguish from other varieties. However, due to the abundance of germplasm resources of the *Castanea*, it is difficult to distinguish all varieties from each other only in terms of phenotypic traits and morphological appearance. For example, both cultivar ‘Wang Zi Tou No.7’ and cultivar ‘Qing Zha’ were clustered into Taxon IV, with no major differences in phenotypic traits such as cone, nut and kernel size, nut shape and nut shell color; cultivar ‘Ba Yue Hong’ and ‘Kui Li’ were both clustered into Taxon II, with similar phenotypic traits and differing only in nut shell color. Comprehensive analysis shows that some of the varieties do not differ significantly in morphological traits and appearance, and it is difficult to identify all varieties accurately only from appearance characteristics and morphological traits.

### 2.2. Polymorphism and Genetic Diversity Analysis

The six SSR loci were tested for neutrality by Ewens–Watterson (neutrality) test, and the observed purity of each locus was determined to be within the 95% confidence interval of the respective expected purity (Appendix A), which were all neutral loci. Moreover, the six pairs of selected chestnut SSR primers had 100% polymorphic loci in 112 Chinese chestnut cultivar resources with no deletion loci, which were suitable for genetic diversity analysis.

The genetic diversity of the 112 Chinese chestnut cultivars was analyzed and evaluated using six chestnut SSR markers (Appendix A). A total of 51 alleles were detected, with an average number of 8.5 alleles per locus. The average number of valid alleles per locus was 3.7061 and the proportion of valid alleles was 43.60%. A total of 91 genotypes were detected, and an average of 15.1667 genotypes were amplified per primer pair. The mean value the of observed heterozygosity was 0.5134, the variances of *He*, *H*, and *PIC* values were similar, and the size of *I* value was approximately 2.15 times larger, with the smallest variation at primer P4 and the largest variation at primer P82, and the average number of variances were 0.7236, 0.7203, 0.6793 and 1.5203, respectively. The results indicated that 112 Chinese chestnut resources had high genetic diversity.

The genetic diversity of 118 *Castanea* variety resources was analyzed and evaluated by adding six resources of other *Castanea* species to 112 Chinese chestnut cultivar resources using six Chinese chestnut SSR markers (Table 2). A total of 58 alleles were detected, with a range of 5–13 alleles and an average number of 9.7 alleles per locus. The number of valid alleles ranged between 2.7602 and 5.4042, with an average of 3.9419 valid alleles per locus; the proportion of valid alleles was 40.78%. A total of 105 genotypes were detected, and the number of genotype species that could be amplified per primer pair ranged from 8 (P4) to 26 (P82), with an average of 17.5 genotypes amplified per primer pair. The observed variation in heterozygosity ranged from 0.1610 to 0.8898, with a mean value of 0.4986. The variances in *He*, *H*, and *PIC* values were similar, and the *I* values were approximately 2.21 times larger than previously measured, with the smallest variation at primer P4 (0.6404, 0.6377, 0.5909, 1.2316, respectively), the largest variation at primer P82 (0.8184, 0.8150, 0.7928, 2.0016, respectively), and the average variation numbers were 0.7390, 0.7359, 0.6985, and 1.6015, respectively. Genetic diversity analysis and evaluation of the 118 *Castanea* resources at the genomic level using six Chinese chestnut SSR markers were generally slightly higher than those containing only Chinese chestnut cultivars, indicating that the six *Castanea* resources of other species also displayed higher genetic diversity and were valuable resources in the *Castanea*.

### 2.3. Population Structure Analysis

Population structure analysis of 118 *Castanea* materials was performed using Structure software, and the optimal grouping was determined by ∆K based on the optimal grouping method. The results showed that ∆K reached its highest peak value when K = 3 (Figure 3a). Therefore, the optimal grouping number was 3. The 118 *Castanea* plant samples were divided into three groups (Figure 3b): Taxon A, Taxon B, and Taxon C (red in the figure represents Taxon A, green represents Taxon B, and blue represents Taxon C). Taxon A consisted of 41 Chinese chestnut species, Taxon B consisted of 39 species of Chinese chestnut, Japanese chestnut (*Castanea* crenata Sieb. et Zucc.), Henry chestnut (*Castanea henryi* (Skam) Rehd. et Wils.), and European chestnut (*Castanea sativa* Mill.), and Taxon C consisted of 38 Chinese chestnut species. According to the major color ratio (*Q* value) in each variety, a *Q* value ≥ 0.6 indicated that the test materials were relatively single in kinship and genetic background, and a *Q* value < 0.6 indicated that the test materials had mixed kinship origin and a more complex genetic background. Of the measured materials, 93.2% had a *Q* value ≥ 0.6, and most of the test materials exhibited a relatively single kinship and genetic background with less gene penetration; 6.8% of the materials with *Q* value < 0.6 were Chinese chestnut cultivars, namely ‘Xin Zhuang No. 2’, ‘Tai Shan Hong Li’, ‘Chui Zhi Li’, ‘Long An No. 1’, ‘Yue You No. 9’, ‘Qing Mao Ruan Zha’, ‘No. 6’, and ‘No. 15’, which each revealed more complex kinship and genetic background. Thus, we can conclude that these materials have complex kinship and genetic backgrounds, and there is some germplasm gene penetration.

The PCoA can directly reflect the relationship between the *Castanea* and plant species. The distance between species is related to genetic similarity. Proximity refers to a high genetic similarity and close kinship. Distance indicates low genetic similarity, long kinship, and large genetic differences. A PCoA of 118 *Castanea* plant species was performed using the first and second principal components as the horizontal and vertical coordinates, respectively (Figure 4), on which 118 *Castanea* plant species were divided into three taxa, namely P1, P2, and P3. Taxa P1 included 49 chestnut plant materials of four species including Chinese chestnut, Japanese chestnut, Henry chestnut, and European chestnut; taxa P2 included 20 Chinese chestnut materials, and Taxon P3 included 49 Chinese chestnut materials. In addition, the distribution of the Chinese chestnut species taxa was scattered, indicating that the tested Chinese chestnut species resources have high diversity.

The NJ clustering diagram (Figure 5) was constructed based on the genetic distance of 118 *Castanea* materials, and it can be seen from the diagram that the 118 *Castanea* materials tested were divided into four taxa, which included Taxon I (yellow), Taxon II (green), Taxon III (blue), and Taxon IV (red). Taxon I included 13 materials, Taxon II included 26 materials, and Taxon III included 15 materials. All materials clustered into Taxon I, Taxon II, and Taxon III were Chinese chestnut species. Taxon IV included Chinese chestnut, Japanese chestnut, Henry chestnut, and European chestnut, with a total of 64 materials.

### 2.4. Establishment of DNA Molecular Identity

It was difficult to identify all materials with a large number of primers in a single pair, and 118 *Castanea* plant materials were identified by combining multiple pairs of primers. By gradually increasing the number of primers, the purpose of identifying all materials was finally achieved. Primer P4 could only identify three parts of the material, but after adding primer P82, 35 parts of the material could be identified and gradually increasing the number of primers up to six pairs of highly polymorphic primers (P4 + P82 + P106 + P108 + P127 + P138) allowed all 118 materials to be distinguished (Table 3).

Complex molecular ID cards must be avoided to facilitate practical production and applications for breeders, producers, and consumers. Therefore, based on the principle of distinguishing the largest species with the least number of primers, for the 118 *Castanea* plant resources, Molecular Identification Card can be constructed using the six primer pairs shown in Table 1. For each pair of primers, each species amplified allele fragment sizes in descending order of 1, 2, 3…9, A, B, C, etc., and in primer order P4, P82, P106, P108, P127, and P138 to form a unique six-digit molecular signature of 118 *Castanea* species (Table 4), which distinguished all materials.

### 2.5. Combining Phenotypic Information with Molecular Information

On the basis of phenotypic trait data and molecular data, the online software forage QR code generator (http://cli.im/ (accessed on 26 February 2023)) was used to encode the desired variety with information such as variety name, germplasm type, botanical classification, phenotypic traits, and string molecular ID card, which well combines phenotypic traits and molecular information. Taking the Chinese chestnut variety ‘Jiu Jia Zhong’ as an example, Figure 6a shows the QR code molecular ID card coded for it. The information shown in Figure 6b,c can be obtained by scanning the QR code, which is very fast and convenient. Analogously, a QR code molecular ID card exclusive to each species can be constructed.

## 3. Discussion

### 3.1. Phenotypic Trait Analysis

In this study, 16 morphological traits were statistically analyzed, among which the coefficient of variation values of five traits, namely, single cone weight, cone shell thickness, single nut weight, stigma length, and single kernel weight, were higher than 30%, which were rich in genetic diversity and had great potential for further selection. The high level of variation in the single cone weight, single nut weight, and single kernel weight, as traits with obvious commercial value [11], will help in the selection of varieties with specific requirements. In addition, two traits with high coefficients of variation, cone shell thickness and stigma length, can be correlated with the development of pistils and the development and maturation of fruits in different varieties for in-depth studies to explore their possible correlation. Fruit shape index had the lowest coefficient of variation and was a relatively stable botanical morphological trait. The mean value of the fruit shape index was 0.81, indicating that the shape of the marginal fruits in the nuts was mostly suborbicular or round, and most of the fruits were square and symmetrical.

Except for the stable phenotypic trait fruit shape index and five phenotypic traits with high coefficients of variation, namely, single cone weight, cone shell thickness, single nut weight, stigma length, and single kernel weight, the values of coefficients of variation for the other 10 phenotypic traits were all in the range of 20% to 30%. Overall, the high values of the coefficients of variation were probably due to the richness of the participating species resources, including materials from four species of *Castanea*, and the large differences in the values of phenotypic traits among the materials of different species, resulting in relatively high overall coefficients of variation.

Phenotypic cluster analysis divided the test material into four groups, with the mean value of single cone weight of 36.01 g for cluster i, which is small and classified as small fruit; the mean value of single cone weight of 82.70 g for cluster ii, which is classified as very large fruit; the mean value of single cone weight of 58.29 g for cluster iii, which is classified as large fruit; and the mean value of single cone weight of 45.56 g for cluster iv, which is medium-sized and classified as medium fruits. The phenotypic traits of different varieties in the same taxon are not very different, and it was difficult to distinguish them from each other in terms of phenotypic traits. Not only were there no significant differences in phenotypic traits among some different varieties, but also the appearance characteristics were very similar, and coupled with the large influence of environmental and other factors on plant phenotypic traits [12], it was more difficult to distinguish and identify each variety by appearance and morphological traits alone.

### 3.2. Genetic Diversity Analysis

SSR molecular markers are among the most commonly used molecular marker technologies, with the advantages of co-dominant inheritance, high stability and polymorphism, and high reproducibility. It is widely used in genetic diversity analysis, kinship determination, variety identification, core germplasm screening, molecular identification, and the construction of various plants [13,14].

The genetic diversity of *Castanea* plants has been studied using SSR molecular markers. For example, Inoue et al. (2009) [15] reported, for the first time, the development of chestnut SSR markers, which achieved effective amplification across species in Japanese and European chestnut using 17 chestnut SSR primers screened for high genetic diversity in all three species of *Castanea* plants, Chinese chestnut, Japanese chestnut, and European chestnut. Nie et al. (2021) [16] screened 18 pairs of primers with high polymorphism from 330 SSR markers to analyze the genetic diversity of 146 Chinese chestnut species and other studies; they measured the mean values of *MAF*, *Ho*, and *PIC* as 0.420, 0.622, and 0.652, respectively, indicating that the 18 SSR markers were highly polymorphic. In this study, six pairs of highly polymorphic primers were screened from 153 pairs of SSR primers that were developed and designed based on the whole Chinese chestnut genome. In 112 Chinese chestnut cultivars, 51 alleles and 91 genotypes were detected, and the mean values of *I*, *Ho*, *He*, *H*, *PIC*, and *MAF* were 1.5203, 0.5134, 0.7236, 0.7203, 0.6793, and 0.4018, respectively, indicating high heterozygosity, polymorphism, and rich genetic diversity. An important reason for the high heterozygosity of Chinese chestnuts is that they are cross-pollinated plants [17]. The SSR molecular markers developed and screened in this study were highly polymorphic, and the rich genetic diversity of the participating cultivars of Chinese chestnut can be explained by the fact that the SSR molecular markers were developed based on high-quality whole-genome sequence data from Chinese chestnut, and that the genomic SSRs exhibited higher genetic diversity [18]. Another possibility is that the involved Chinese chestnut species possessed more resources and complex genetic information. To address the issue of low resolution while performing polyacrylamide gel electrophoresis, this study used fluorescent markers and capillary electrophoretic methods to read and record data, which greatly improved the accuracy of allelic discrimination of loci and the detection of target fragment sizes.

In addition, six resources of three *Castanea* species, Japanese chestnut, Henry chestnut, and European chestnut, were added to 112 Chinese chestnut species. The SSR molecular markers developed from the Chinese chestnut genome were successfully and efficiently amplified across species in Japanese chestnut, Henry chestnut, and European chestnut, indicating that the developed SSR molecular markers were highly versatile in the *Castanea* and can be used for genetic diversity analysis, species identification, and molecular ID construction of the *Castanea*.

### 3.3. Population Structure Analysis

Population structure, NJ cluster analysis, and principal coordinate analysis are effective means to study the genetic diversity, kinship, and genetic background of plant germplasm resources and are critical bases for protecting and utilizing plant germplasm resources. Many studies have been conducted on the population structure of plants [19,20,21,22,23,24], and related studies have also been reported in *Castanea* plants. For example, Jiang et al. (2017) [25] analyzed the genetic structure of 95 Chinese chestnut cultivars using 41 SSR loci and classified 95 Chinese chestnut cultivars into three taxa. Nishio et al. (2014) [26] classified 60 Japanese chestnut cultivars into two major clusters and three major clusters by hierarchical clustering and Bayesian clustering analysis, respectively, and the results of both analyses showed some similarity. Pereira-Lorenzo et al. (2017) [27] used 24 highly polymorphic SSRs to analyze the population structure of 132 European chestnut cultivars, identifying two major clusters corresponding to the Spanish and Italian cultivar taxa and showing higher genetic diversity. While the above studies only analyzed the resources of individual species of the *Castanea* for structural analysis, the present study involved Chinese chestnut, Japanese chestnut, Henry chestnut, and European chestnut.

In this study, six pairs of highly polymorphic SSR markers were used to analyze the structure of 118 *Castanea*. A total of 118 *Castanea* plants were classified using three analytical methods: structural analysis, NJ clustering analysis, and PCoA analysis. The classification results of the three methods were highly consistent. Taxon A of the structure analysis results contained all the materials in Taxon I of the NJ cluster analysis except for the Chinese chestnut cultivar ‘Shi Men Zao Shuo’. All the remaining 32 materials in Taxon A except for the nine materials of ‘Jiao Zha’, ‘Jian Ding You Li’, ‘Yue Yao No. 8’, ‘Jie Jie Hong’, ‘Long An No. 1’, ‘Mei Gui Hong’, ‘Qing Mao Ruan Zha’, ‘ZA’, and ‘No. 15′ were clustered into Taxon P3 of the PCoA analysis. The remaining 36 materials in the B Taxon of the structure analysis were clustered into the P1 Taxon of the PCoA analysis, except for ‘Mao Pu’, ‘Chui Zhi Li’, and ‘No. 6’; the remaining 48 materials in the P1 Taxon were clustered into the IV Taxon of the NJ clustering analysis, except for ‘Shi Men Zao Shuo’. The P3 Taxon of the PCoA analysis included all materials from Taxon II of the NJ clustering analysis; six materials, including Japanese chestnut, Henry chestnut, and European chestnut, were grouped into the same taxon by structure analysis, NJ cluster analysis, or PCoA analysis. Although there was a high degree of agreement and similarity in the classification results among structure analysis, NJ clustering analysis, and PCoA analysis, there were also differences, with different clusters on select branches. The structural analysis and PCoA analysis divided the 118 materials into three broad categories, and the NJ clustering analysis divided the 118 materials into four broad categories.

In the population structure analysis, 6.8% of the materials had a *Q* value of less than 0.6, indicating a certain frequency of gene exchange and infiltration among the participating material species. However, due to the lack of clarity regarding the origin and genetic background of certain resources, it was not possible to link taxonomy to geographic origin for analysis. In future studies, we will attempt to track the origin and genetic background information of each material to explore the genetic structure, genetic diversity, and affinity between different geographic resources.

### 3.4. Establishment of DNA Molecular Identity

Molecular identification is an effective method to solve the problems of species mixing, tautology, and synonyms and is an important guide for species identification. To date, many studies on molecular ID have been reported [28,29,30,31,32,33], but there are few reports on the molecular ID construction in *Castanea* plants. Currently, however, there are many substandard phenomena, which are not conducive to market development. As an example, consider the fried Chinese chestnut market in Nanjing, Jiangsu Province, the market survey revealed that most of Nanjing’s fried Chinese chestnuts are made from Yanshan Chinese chestnuts produced in Qianxi, Hebei Province, which are soft and sweet and liked by consumers. However, most operators and consumers of fried Chinese chestnut are not able to distinguish between Yanshan Chinese chestnut and other varieties. The presence of mixed varieties in the market and the phenomenon of using the substandard as a good is endangering the legitimate rights and interests of both operators and consumers. Therefore, it will be necessary and meaningful to construct molecular ID cards to standardize the management of *Castanea* plants.

Nie et al. (2021) [16] constructed fingerprint profiles of 146 Chinese chestnut materials using seven core markers selected, which enabled rapid and effective identification of different Chinese chestnut germplasm resources. Liu et al. (2017) [34] screened five pairs of primer combinations and used the core primer plus band type combination method to construct fingerprint profiles of 33 ancient Chinese chestnut trees from the Ming and Qing dynasties, which each made important contributions to the research and conservation of ancient Chinese chestnut trees. In this study, we constructed 118 exclusive molecular IDs of the *Castanea* plants using six pairs of SSR core primers, and were able to completely distinguish 118 materials, which not only provided a reference for the identification and authentication of the *Castanea* plants, but will also contribute to the development of the *Castanea* plant market.

### 3.5. Phenotype and Molecular Binding Analysis

Fruit phenotypic morphological traits have significant economic value and discriminatory ability, and are widely used in the identification of different plant varieties. However, due to environmental factors and plant growth cycles, phenotypic traits are somewhat unstable, and some varieties have very similar phenotypic traits. Therefore, it is difficult to identify all varieties using phenotypic traits alone. At present, rapid development and increasing sophistication of molecular marker technologies will help overcome the limitations of phenotypic traits. The combination of phenotypic and molecular-based methods will help to identify varieties more accurately and comprehensively [35,36,37,38]. In this study, the phenotypic cluster analysis divided 118 materials into four groups, and the phenotypic traits of the varieties differed significantly among the groups; however, the phenotypic traits of the varieties within each group did not differ significantly. Therefore, it was difficult to identify all materials using only phenotypic traits. In this study, the two-dimensional molecular identity of Chinese chestnut cultivar ‘Jiu Jia Zhong’ was constructed as an example, which contains both phenotypic trait information and DNA molecular information, and the construction of the two-dimensional molecular identity will help to identify *Castanea* plant resources in detail and comprehensively.

In addition, phenotypic trait clustering analysis and genetic distance clustering analysis showed that although the 118 materials were grouped into four groups by the two methods, the clustering results were different between the two methods. Genetic distance of varieties with similar phenotypic traits may be distant, for example, the cultivar ‘Duan Zha’ and the cultivar ‘Gui Hua Xiang’; phenotypic traits of varieties with similar genetic distance may have large differences, for example, the cultivar ‘Shi Men Zao Shuo’ and the cultivar ‘Cen Kou Da Li’. However, some of the materials were grouped together in both phenotypic and genetic distance clusters, for example, the cultivar ‘Chu Shu Hong’ and the cultivar ‘Bo Ke Chi Li’, the cultivar ‘Shu He No.1′ and the cultivar ‘Shu He No.7’, indicating that their genes and relatives may be very similar.

## 4. Materials and Methods

### 4.1. Plant Materials

From April to October 2021, 118 materials (Appendix A) from 4 species of the *Castanea* were collected from Jiangsu Province, China. We collected 10–15 young leaves, marked them, placed them in self-sealing bags, brought back to the laboratory in ice boxes, and stored at −20 °C in a refrigerator.

### 4.2. Phenotype Data Measurement

The phenotypic traits of different varieties of the *Castanea* germplasm resources were measured for three consecutive years from 2020 to 2022, and the appearance characteristics were photographed in a varietal box plot. A vernier caliper was used to measure the phenotypic data of 16 traits, including Single cone weight, Cone diameter, Cone length, Cone thickness, Spine length, Cone shell thickness, Single nut weight, Nut diameter, Nut length, Nut thickness, Fruit shape index, Stigma length, Single kernel weight, Kernel diameter, Kernel length, Kernel thickness. Each species has at least 3 replications. The maximum, minimum, mean, standard deviation and coefficient of variation were calculated for each trait. Cluster analysis of phenotypic traits was performed using RStudio software.

### 4.3. DNA Extraction and PCR Amplification

Leaf genomic DNA was extracted using a modified cetyltrimethylammonium bromide (CTAB) method [39]. Two percent mercaptoethanol was added prior to use of the extract, and DNA quality and concentration were detected by Nano-100 instrument, diluted to 50 ng/μL, and stored at −20 °C for rescue.

Six pairs of primers with high polymorphism (*PIC* > 0.5) were selected from 153 pairs of primers developed and synthesized based on published chestnut genome sequence data [40] (Table 5). The fluorescence modification was performed at the 5′ end of the upstream primer of each pair.

Fluorescent SSR-PCR amplification was performed using a 15 μL reaction system: 7.5 μL 2 × TSINGKE Master Mix (Tsingke, Nianjing, China), 1 μL template DNA (50 ng/μL), 0.3 μL of 10 μM fluorescence-modified upstream primer, 0.3 μL of 10 μM downstream primer, and 5.9 μL ddH_2_O to produce 15 μL reaction solution for PCR.

The PCR amplification reaction procedure was as follows: pre-denaturation at 94 °C for 3 min; denaturation at 94 °C for 30 s, annealing at 57 °C for 30 s, extension at 72 °C for 45 s and a total of 35 cycles; full extension at 72 °C for 10 min; and storage at −20 °C protected from light. Capillary electrophoresis detection was performed using an ABI 3730 XL DNA Sequencer (Applied Biosystems, Foster City, CA, USA), and Gene Mapper 4.1 software was used for data collation and image analysis.

### 4.4. Data Analysis

The data format was converted using DataFormater software [41], and POPGENE software version 1.32 [42,43] was used to calculate the number of alleles (*Na*), the number of effective alleles (*Ne*), Shannon’s information index (*I*), observed heterozygosity (*Ho*), expected heterozygosity (*He*), Nei’s genetic diversity index (*H*) [44], and Ewens–Watterson (neutrality) test. Polymorphism information content (*PIC*), Number of genotypes (*NG*) and master allele frequency (*MAF*) were calculated using PowerMarker V3.25 [45]. Additionally, calculating allele frequencies and genetic distances, performing NJ clustering analysis, and the NJ clustering analysis data were embellished with iTOL website (https://itol.embl.de/ accessed on 26 February 2023).

Population genetic structure analysis was performed using Structure software [46,47,48], and 15 replicate runs were performed with K values ranging from 2 to 8. The length of the burn-in period was set at 10,000, and the number of Markov chain Monte Carlo (MCMC) repeats after burn-in was set at 100,000 to estimate individual admixture proportions (*Q*). Evanno’s method [49] and the online software Structure Harvester [50] were used to determine the optimal grouping K values [51,52,53]. Principal coordinate analysis (PCoA) was performed using GenAlex 6.41 software [54,55].

The allelic genotypic fragments amplified by each primer pair were ranked from smallest to largest and numbered 1, 2, 3…9, A, B, C, etc., and the primer pair used P4, P82, P106, P108, P127, and P138. The sequence was arranged to construct fingerprint feature codes to obtain the molecular information of each material. The botanical taxonomic information of the materials was combined with molecular information using 2D code online generation technology (https://cli.im/ accessed on 26 February 2023) to generate a two-dimensional code molecular identity card.

## 5. Conclusions

In this study, 16 phenotypic traits were measured and analyzed; among them, the coefficient of variation of five phenotypic traits such as single cone weight was greater than 30%, and the coefficient of variation of fruit shape index was the smallest, which was a relatively stable trait. The genetic diversity and population structure of 118 *Castanea* plants were analyzed using SSR molecular markers; the results showed that the test materials had rich genetic diversity. The molecular ID cards of 118 *Castanea* plant materials were successfully constructed using six pairs of core SSR primers, which can completely differentiate all participating materials and have important implications for the conservation, utilization, and management of *Castanea* plant resources.

## Figures and Tables

**Figure 1 plants-12-01438-f001:**
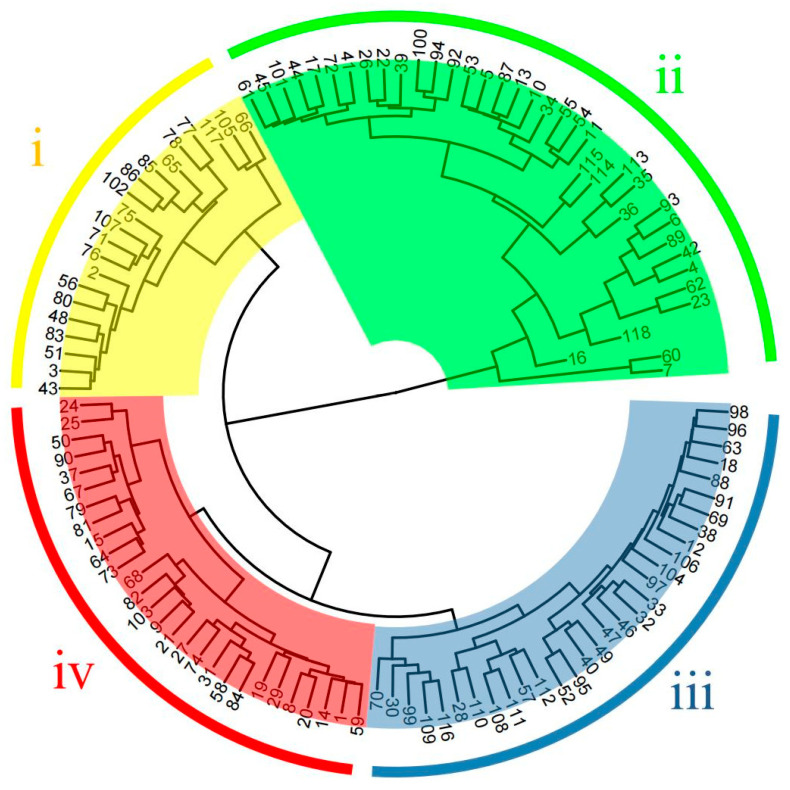
Cluster analysis of phenotypic traits in 118 materials. The labels in the figure represent the material number (corresponding to Appendix A).

**Figure 2 plants-12-01438-f002:**
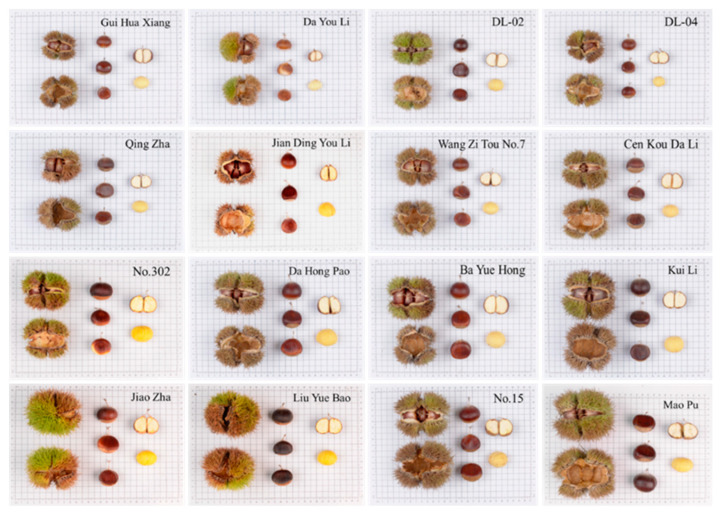
Cone seed atlas of different cultivars of the *Castanea* germplasm resources.

**Figure 3 plants-12-01438-f003:**
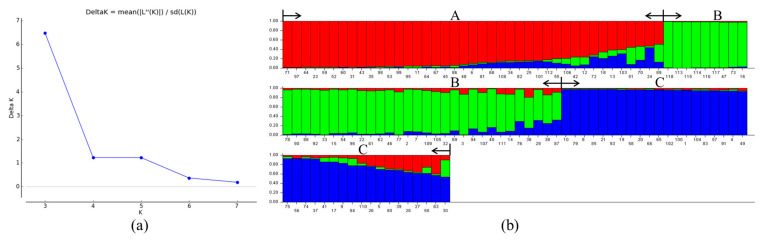
Optimal K-value according to maximum Delta K (**a**) and population structure analysis of 118 Castanea plant materials (**b**). The optimum number of groups is 3, dividing the 118 ingredients into three groups (A, red; B, green; C, blue). The labels on the bottom of (**b**) represent the material number (corresponding to Appendix A).

**Figure 4 plants-12-01438-f004:**
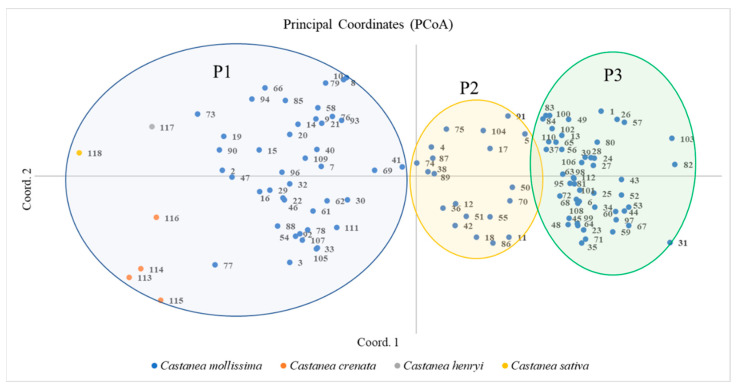
PCoA of 118 *Castanea* plant materials. The labels in the figure represent the material number (corresponding to Appendix A).

**Figure 5 plants-12-01438-f005:**
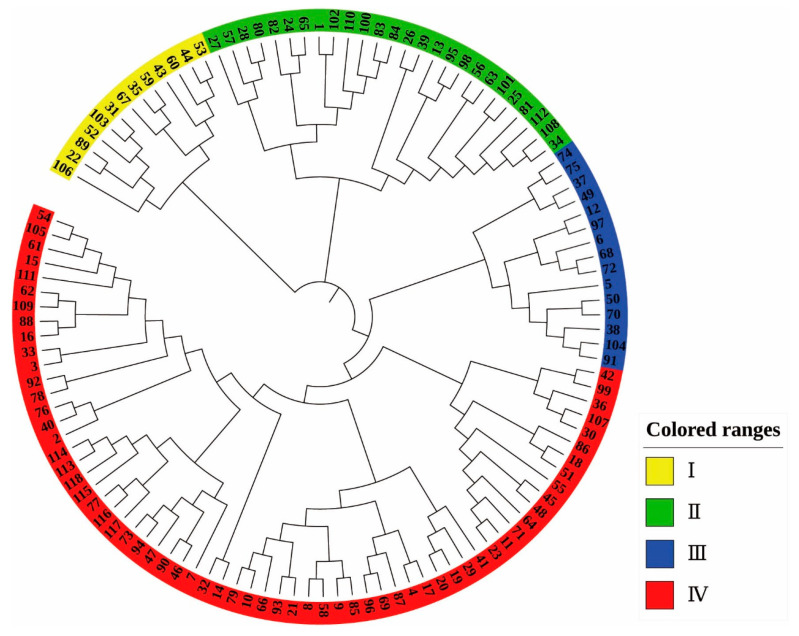
NJ clustering analysis of 118 Castanea plant materials. The labels in the figure represent the material number (corresponding to Appendix A).

**Figure 6 plants-12-01438-f006:**
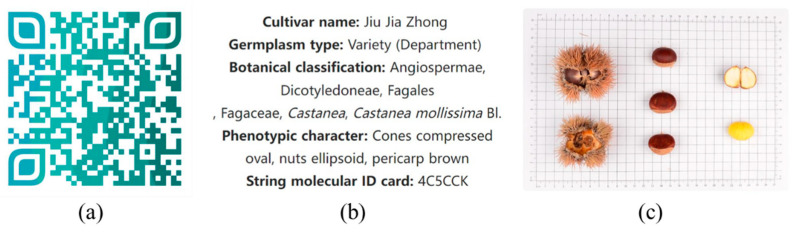
Example of QR code molecular ID card. (**a**) QR code molecular ID card of Chinese chestnut variety ‘Jiu Jia Zhong’; (**b**,**c**) display of the contents of the scan code.

**Table 1 plants-12-01438-t001:** Descriptive statistical analysis of phenotypic traits used for characterization of the *Castanea* varieties.

Phenotypic Traits (Unit)	Mean	Max	Min	SD	CV (%)
Single cone weight (g)	58.05	151.32	4.00	22.81	39.30%
Cone diameter (mm)	71.84	140.02	31.67	17.76	24.72%
Cone length (mm)	60.64	155.01	29.33	17.14	28.27%
Cone thickness (mm)	56.60	115.38	26.67	15.06	26.61%
Spine length (mm)	12.24	24.31	6.07	2.93	23.94%
Cone shell thickness (mm)	2.40	6.37	0.79	0.89	37.08%
Single nut weight (g)	10.45	37.67	2.00	5.19	49.67%
Nut diameter (mm)	29.89	56.73	11.00	6.79	22.72%
Nut length (mm)	24.06	46.42	11.33	5.26	21.86%
Nut thickness (mm)	21.55	47.59	11.89	5.64	26.17%
Fruit shape index	0.81	1.03	0.55	0.09	11.11%
Stigma length (mm)	11.03	28.86	2.70	3.82	34.63%
Single kernel weight (g)	8.33	36.17	1.70	5.03	60.38%
Kernel diameter (mm)	27.82	52.07	9.87	6.61	23.76%
Kernel length (mm)	21.93	42.49	10.10	4.86	22.16%
Kernel thickness (mm)	18.22	151.32	8.02	5.19	28.49%

Note: SD: standard deviation; CV: coefficient of variation.

**Table 2 plants-12-01438-t002:** Genetic diversity parameters of 118 *Castanea* plant materials.

Primer Name	*Na*	*NG*	*Ne*	*I*	*Ho*	*He*	*H*	*PIC*	*MAF*
P4	5	8	2.7602	1.2316	0.1610	0.6404	0.6377	0.5909	0.5339
P82	13	26	5.4042	2.0016	0.5254	0.8184	0.8150	0.7928	0.2966
P106	12	23	4.2003	1.7045	0.8898	0.7652	0.7619	0.7265	0.3517
P108	9	14	3.6012	1.5635	0.2373	0.7254	0.7223	0.6894	0.4534
P127	7	13	3.6792	1.4494	0.5763	0.7313	0.7282	0.6790	0.3305
P138	12	21	4.0063	1.6584	0.6017	0.7536	0.7504	0.7124	0.3602
Mean	9.6667	17.5	3.9419	1.6015	0.4986	0.7390	0.7359	0.6985	0.3877

**Table 3 plants-12-01438-t003:** Materials distinguished using a gradually increasing number of primer pairs.

Primer Combination	Materials Can Be Distinguished	Subtotal	Total
P4	115, 117, 118	3	3
P4 + P82	2, 3, 7, 11, 14, 22, 26, 30, 32, 33, 38, 39, 40, 45, 55, 61, 64, 66, 73, 76, 77, 78, 83, 84, 85, 95, 97, 100, 103, 113, 114, 116	32	35
P4 + P82 + P106	4, 5, 6, 8, 12, 15, 17, 18, 19, 20, 24, 29, 31, 41, 42, 43, 48, 51, 58, 67, 70, 71, 81, 82, 86, 87, 89, 92, 93, 98, 99, 106, 107, 109, 110	35	70
P4 + P82 + P106 + P108	9, 10, 13, 21, 23, 25, 36, 46, 47, 49, 50, 52, 56, 57, 65, 69, 74, 75, 79, 88, 90, 94, 96, 111, 112	25	95
P4 + P82 + P106 + P108 + P127	1, 27,28, 35, 37, 44, 53, 54, 59, 60, 63, 80, 91, 101, 102, 104, 105	17	112
P4 + P82 + P106 + P108 + P127 + P138	16, 34, 62, 68, 72, 108	6	118

Note: The labels in the table represent the number of materials used (corresponding to Appendix A).

**Table 4 plants-12-01438-t004:** Six-digit molecular ID codes of 118 *Castanea* plant materials.

Code	Name of Variety	Molecular Identity Code	Code	Name of Variety	Molecular Identity Code
1	Qing Zha	2CHCCK	60	Shen Ci Da Ban Li	2E784H
2	Shu He No.1	6O9A3A	61	Xiao Jing Tie Li	65J92D
3	Shu He No.7	6Q983A	62	Zhong Chi Li	6C942K
4	Shu He No.10	37574D	63	Yue You No.9	27793G
5	Da Di Qing	2P5A8G	64	Te Zao	21783F
6	Da Hong Pao	2P788G	65	Wu Mao Tie Li	2CHD4D
7	Da Gong Shu No.4	1C844D	66	Duan Zhi Li	495C4B
8	You Zao No.1	474ACH	67	Ban Li Zi	2HM888
9	Zao Li Zi	4C5B6D	68	Hong Ming Jian	23788G
10	Jiu Jia Zhong	4C5CCK	69	Hu Bei You Li	3B513B
11	Jiao Zha	3G783K	70	Qing Mao Ruan Zha	23584A
12	Jian Ding You Li	3Q58CC	71	CKD	2HD83F
13	Dong Wang Ming Li	2772DC	72	DL-01	23788C
14	Hong Li	1G4C8G	73	DL-02	136C5E
15	Xiao Luan Shi	6CJCB2	74	DL-03	3O7D7C
16	Huang Qian Zhong Wan	6C942B	75	DL-04	3O7C8C
17	Lian Hua Li	225727	76	MJH	83AA4C
18	Yue You No.8	3AH87H	77	W4	1EF82I
19	Mi Feng Qiu	47ID7G	78	W5	6PJ844
20	Wang Zi Tou No.7	475A7H	79	XHC	425CCD
21	Gao Yuan No.1	42564D	80	XBC	2O79CH
22	Shi Men Zao Shuo	5EE963	81	YBH	22742H
23	Er Shui Zao	2G783F	82	YML	2QMA2K
24	Xin Zhuang No.2	2GKD4D	83	Y46	287C4B
25	Wei Hai Zao Shu	27784H	84	Y47	2JDC4B
26	Chu Shu Hong	2A7ACH	85	ZMZ	4O4D8D
27	Yan Hong	27798K	86	ZA	3QH83H
28	Xiao Xue	2O794K	87	No.6	3A534D
29	Duan Zha	62JB7H	88	No.9	6C983B
30	Tai Shan Hong Li	57588D	89	No.15	3EB75D
31	Gui Hua Xiang	2EM86F	90	No.17	1O523C
32	Yan Shan Zao Feng	6E9C7H	91	No.18	2O5C2G
33	Yang Guang No.2	6M982L	92	No.101	7CF83A
34	Bo Ke Chi Li	2C782K	93	102B	428D4K
35	Huang Li Pu	2HH87H	94	No.105	1O6C4G
36	Mao Pu	3B588D	95	No.108	24D27H
37	Xin Yi You Li	2CHC7H	96	No.203	6C533C
38	Chui Zhi Li	3P5948	97	No.207	26789G
39	Nian Di Ban	2N7A3H	98	No.213	2Q727G
40	Yue Xi No.2	8LAA3L	99	No.302	2E588K
41	Wu Ke Li	37FA3H	100	No.1059	297C4B
42	Jie Jie Hong	3E588D	101	No.1061	2C784G
43	Cen Kou Da Li	2GD4CH	102	No.1504	2CHC4D
44	Liu Yue Bao	2E78AH	103	8017	2FMC5K
45	Ba Yue Hong	2D783F	104	Liu He Hong Li	2O5C3C
46	Hua Gai	1O588C	105	-	67J83K
47	Huang Qian Wu Hua	1O648I	106	-	2EKA8I
48	Su Cheng Da Li	27H83F	107	-	62587D
49	Da You Li	2C7C8G	108	-	2C782H
50	Gui Xuan 72-1	2O584B	109	-	67924H
51	Long An No.1	37N47K	110	Gan Yu No.1	22H54C
52	He Bei Zun Yu	2G745K	111	-	6C582K
53	Kui Li	2E78CJ	112	-	2O742H
54	Jiu Yue Han	67J82D	113	Yin Ji	1I2879
55	Mei Gui Hong	3C783K	114	-	1J3879
56	Wang Jie Gen	277B3G	115	-	6I1871
57	Chen Guo You Li	2C76CK	116	-	1KG429
58	Shuang He Da Hong Pao	4C4A2K	117	-	1JCEC5
59	Er Xin Zao	2HH89H	118	-	1IL516

Note: ‘-’ represent the material is wild resource, does not have variety name.

**Table 5 plants-12-01438-t005:** Primer Information.

Primer	Primer Sequence (5′−3′)	Chromosome No.	Allele Size	Fluorescent Dyes
P4	F-GATTGTGCAACAACACCTGC R-CAACCCTGCCAAGAGATTGT	1	173–181	TET
P82	F-CTCTGGGTTTACCTTGGGCT R-CGGGCTGAGTTTGGTTAAAA	5	155–189	TAMRA
P106	F-GAGCAAGCTGCTACCCTGAC R-CGGTCTCAGATTTCAGGCTC	7	163–181	Fam
P108	F-TCGCATGTTGTCCTTTACGA R-TCTCCGAGTTCTCCCTCTGA	8	176–189	Hex
P127	F-TTCCAATGGACCAATACCGT R-CCCACATGGCCTCATTCTAT	9	184–250	ROX
P138	F-CGAGGGAATTATGAGGGTTTT R-CAAAATGCTCAAGGGGGTAA	11	174–201	Fam

## Data Availability

The data that support the findings of this study are available from the corresponding author upon reasonable request.

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
