# Peer review of "Genetic Relationships of 118 Castanea Specific Germplasms and Construction of Their Molecular ID Based on Morphological Characteristics and SSR Markers"

_plants, 2023, doi:10.3390/plants12071438_

Round 1

Reviewer 1 Report

The results of this study are expected to provide a molecular basis for understanding the genetic diversity and population structure of germplasm resources of the genus Castanea, and to be a reference for resolving the problem of homonymy and agreement. Therefore, I suggest that this paper can be accepted for publication in Plants, but need minor revisions.

The comments or suggestions are as below:

- Lines 13, 22, 25: Should identify abbreviation of SSR, NJ, PCoA and ID on the first use.

- Lines 230-233, Table 4: In order to help readers understand, it is necessary to explain the morphological characteristics of 118 genera Castanea plant materials classified by Six-digit molecular ID codes.

- Lines 484, 5. Conclusions: The present conclusions are too brief. Therefore, a more detailed explanation of the findings in this study is needed.

Author Response

Point 1: - Lines 13, 22, 25: Should identify abbreviation of SSR, NJ, PCoA and ID on the first use.

Response 1: Thank you very much for the valuable suggestions of the expert teacher, I have completed the full names of these abbreviations where they appear for the first time in the article.

Point 2: - Lines 230-233, Table 4: In order to help readers understand, it is necessary to explain the morphological characteristics of 118 genera Castanea plant materials classified by Six-digit molecular ID codes.

Response 2: Thank you very much for the valuable suggestions made by the expert teacher, because the table in the text can present limited content, so we put the phenotypic trait characteristic data of the test material in the attached table. We performed cluster analysis based on phenotypic trait data and molecular marker data on the test materials in the text, and the test materials were classified in both cluster analysis plots, and we also performed comparative and combined analysis of phenotype and molecular classification. Thank you very much for the valuable suggestions of our expert teachers, and we will strive to present information concisely and comprehensively in our future study and work.

Point 3: - Lines 484, 5. Conclusions: The present conclusions are too brief. Therefore, a more detailed explanation of the findings in this study is needed.

Response 3: Thank you very much for the valuable suggestions made by the expert teacher, I revised and supplemented the conclusion in the text according to the advice of the expert teacher. Many thanks to the expert teacher, because there are valuable suggestions from expert teachers to make the article better.

Reviewer 2 Report

In this manuscript, Bai et al. present their work on deciphering the genetic relationships of more than 100 varieties of the genus Castanea by analyzing SSR markers. They also combine genetic with phenotypic data and generate molecular IDs for each cultivar.

The article is decently written, introducing the subject adequately. The data are appropriately acquired and clearly presented, but, in general, the study does not shine for its novelty or breakthrough findings. Nevertheless, its results and conclusions might prove helpful for several research communities.

Specific comments:

1. L. 122: Please, replace "Ewens-Watterson" with something like "Ewens-Watterson (neutrality) test" for better reading.

2. L. 356-358: It is not clear whether PCoA analysis yields three groups, instead of, say, four, since it seems that the largest amount of variability lies on the first (horizontal) axis.

3. With respect to the previous comment, it would be helpful to use the same color coding for population grouping in Figures 3 & 4 (and maybe 5).

4. Please provide statistical support for NJ clades in Figure 5.

5. L. 315 - 316: Please rephrase for better English.

6. Pls replace the phrase "genera Castanea", which is repeated more than 50 times throughout the text, with "genus Castanea", or simply "Castanea". "Genera" is the plural form of the word "Genus".

Author Response

Point 1: L. 122: Please, replace "Ewens-Watterson" with something like "Ewens-Watterson (neutrality) test" for better reading.

Response 1: Thank you very much for the valuable suggestions made by the expert teacher, I have revised the text accordingly based on the advice of the expert teacher.

Point 2: L. 356-358: It is not clear whether PCoA analysis yields three groups, instead of, say, four, since it seems that the largest amount of variability lies on the first (horizontal) axis.

Response 2: Thanks to the valuable suggestions of expert teacher, we have made a general classification in PCoA analysis, and we will analyze it in more detail in future study and work. Thanks again to expert teacher for your valuable advice, which is very helpful to us.

Point 3: With respect to the previous comment, it would be helpful to use the same color coding for population grouping in Figures 3 & 4 (and maybe 5).

Response 3: Thank you very much for the valuable advice of the expert teacher, I think it is very useful. However, because Figures 3, 4, and 5 are plotted using their respective analysis software, there may be color differences between different software. However, I have tried to keep the color of the result map of these three analysis methods more consistent, mainly using the four colors of red, blue, green and yellow, because the PCoA analysis result map needs to reflect the material serial number in the figure clearly, so the corresponding color with better transparency is used, which makes the color depth may be quite different from the other two pictures. But each image sheds light on its own content and meaning. Thanks again to the expert teacher for their valuable advice, which is very helpful for my future research and study.

Point 4: Please provide statistical support for NJ clades in Figure 5.

Response 4: Thank you very much expert teacher, I will provide statistical support for the NJ clade in the attachment.

Point 5: L. 315 - 316: Please rephrase for better English.

Response 5: Thank you very much for the valuable advice of the expert teacher, I have revised the wording of the corresponding part of the text to try to make the expression more accurate and better.

Point 6: Pls replace the phrase "genera Castanea", which is repeated more than 50 times throughout the text, with "genus Castanea", or simply "Castanea". "Genera" is the plural form of the word "Genus".

Response 6: Thank you very much for the valuable suggestions of the expert teacher, I changed the phrase "genera Castanea" to "Castanea" in 56 places in the text according to the advice of the expert teacher. Thanks again to the expert teacher, your valuable suggestions make the article better.

Reviewer 3 Report

Number of markers are very less this information will not help to extend the current knowledge of researchers 

Author Response

Point 1: give some info like family, chromosome number, pollination behaviour, which part is economicallly important.

Response 1: Thank you very much for the valuable suggestions made by the expert teacher, I added the corresponding information to the article according to the advice of the expert teacher, hoping to make the article better.

Point 2: how many replication for collection of data

Response 2: Thank you very much for the valuable advice of the expert teacher, we took at least 3 repetitions of each variety, and I added this in the corresponding place in the text according to the valuable advice of the expert teacher. Thanks again to the expert teacher, because the valuable advice of the expert teacher made the article better.

Point 3: number of markers are very less, this will not give correct and relaiable information on molecular tag

Response 3: Thank you very much for the valuable advice of the expert teacher, because we want to use as few primers as possible to identify as many varieties as possible, avoid excessive molecular ID information, and obtain an efficient, concise and convenient molecular ID code for application. In this process, we selected several pairs of primers with high polymorphism for study. Relatively few primers were selected for analysis, did not cover the entire genome, and may vary. However, the primers we screened for analysis were all highly polymorphic primers, which were representative, and we hoped to minimize the differences to a certain extent. Thanks again for the valuable advice of the expert teacher, which will also be very useful for my future study and work.

Point 4: there is no mean to perform structure analysius when number of markers are not coversing genome

Response 4: Thank you very much for the valuable advice of the expert teacher, because we want to use as few primers as possible to identify as many varieties as possible, avoid excessive molecular ID information, and obtain an efficient, concise and convenient molecular ID code for application. In this process, we selected several pairs of primers with high polymorphism for study. Relatively few primers were selected for analysis, did not cover the entire genome, and may vary. However, the primers we screened for analysis were all highly polymorphic primers, which were representative, and we hoped to minimize the differences to a certain extent. Thanks again for the valuable advice of the expert teacher, which will also be very useful for my future study and work.

Round 2

Reviewer 3 Report

Authors resolved the query raise in original version.